# Sequential Behavior of Broiler Chickens in Enriched Environments under Varying Thermal Conditions Using the Generalized Sequential Pattern Algorithm: A Proof of Concept

**DOI:** 10.3390/ani14132010

**Published:** 2024-07-08

**Authors:** Juliana Maria Massari, Daniella Jorge de Moura, Irenilza de Alencar Nääs, Danilo Florentino Pereira, Stanley Robson de Medeiros Oliveira, Tatiane Branco, Juliana de Souza Granja Barros

**Affiliations:** 1School of Agricultural Engineering, State University of Campinas, 501 Candido Rondon Avenue, Campinas 13083-875, SP, Brazil; jujumassari@gmail.com (J.M.M.); stanley.oliveira@embrapa.br (S.R.d.M.O.); jgbarros@unicamp.br (J.d.S.G.B.); 2Graduate Program in Production Engineering, Paulista University, R. Dr. Bacelar 1212, São Paulo 04026-002, SP, Brazil; irenilza.naas@docente.unip.br; 3School of Sciences and Engineering, Department of Management, Development and Technology, São Paulo State University (UNESP), Tupã 17602-496, SP, Brazil; danilo.florentino@unesp.br; 4Embrapa Digital Agriculture, State University of Campinas, 209 Andre Tosello, Campinas 13083-886, SP, Brazil; 5Agricultural and Forestry/Animal Scientist Analyst, Secretariat of Agriculture, Livestock, Sustainable Production, and Irrigation—SEAPI, 1384 Getulio Vargas Avenue, Porto Alegre 90820-150, RS, Brazil; tatibranco91@gmail.com

**Keywords:** animal behavior, animal welfare, data mining, environment, machine learning, precision livestock farming

## Abstract

**Simple Summary:**

Animal production workers routinely use postural and behavioral patterns to assess thermal comfort and assist in environmental management. This study investigated the behavior sequences of broiler chickens housed in enriched environments subjected to thermal comfort and heat stress using the Generalized Sequential Pattern (GSP) data mining algorithm. The comfort temperature resulted in a greater number of behavioral patterns compared to heat stress. Heat is a limiting factor for the occurrence of patterns, restricting movement and decreasing the group’s activity level. The GSP identified temporal correlations between heat stress and the behavior of broiler chickens in enriched environments and can be an efficient alternative for monitoring and possibly diagnosing heat stress.

**Abstract:**

Behavior analysis is a widely used non-invasive tool in the practical production routine, as the animal acts as a biosensor capable of reflecting its degree of adaptation and discomfort to some environmental challenge. Conventional statistics use occurrence data for behavioral evaluation and well-being estimation, disregarding the temporal sequence of events. The Generalized Sequential Pattern (GSP) algorithm is a data mining method that identifies recurrent sequences that exceed a user-specified support threshold, the potential of which has not yet been investigated for broiler chickens in enriched environments. Enrichment aims to increase environmental complexity with promising effects on animal welfare, stimulating priority behaviors and potentially reducing the deleterious effects of heat stress. The objective here was to validate the application of the GSP algorithm to identify temporal correlations between heat stress and the behavior of broiler chickens in enriched environments through a proof of concept. Video image collection was carried out automatically for 48 continuous hours, analyzing a continuous period of seven hours, from 12:00 PM to 6:00 PM, during two consecutive days of tests for chickens housed in enriched and non-enriched environments under comfort and stress temperatures. Chickens at the comfort temperature showed high motivation to perform the behaviors of preening (P), foraging (F), lying down (Ld), eating (E), and walking (W); the sequences <{Ld,P}>; <{Ld,F}>; <{P,F,P}>; <{Ld,P,F}>; and <{E,W,F}> were the only ones observed in both treatments. All other sequential patterns (comfort and stress) were distinct, suggesting that environmental enrichment alters the behavioral pattern of broiler chickens. Heat stress drastically reduced the sequential patterns found at the 20% threshold level in the tested environments. The behavior of lying laterally “Ll” is a strong indicator of heat stress in broilers and was only frequent in the non-enriched environment, which may suggest that environmental enrichment provides the animal with better opportunities to adapt to stress-inducing challenges, such as heat.

## 1. Introduction

Chicken meat is distinguished by being a low-cost, high-quality source of protein, and it is considered one of the most environmentally friendly meats to produce [1]. In traditional livestock farming, decision-making criteria primarily rely on the producer’s experiential and empirical knowledge [2]. In the routine management practices of farmers, particular attention is paid to broiler chickens’ behavioral activity and spatial occupancy patterns during visual inspection. This process demands a considerable amount of time [3]. In precision livestock farming (PLF), decision-making relies on quantitative data obtained through process engineering principles and techniques. Such an approach enables the automatic monitoring, modeling, and management of animal production [4]. Therefore, in precision livestock farming (PLF), technologies enable the continuous collection of behavioral and physiological data at the individual level [5], allowing for the automated and real-time monitoring of critical indicators of herd behavior.

The substantial collection and storage of data in smart agricultural production has presented a new challenge: how can this large volume of data be processed to derive actionable insights for optimizing animal production? This situation introduces an additional layer of complexity, which can be addressed by applying machine learning and data mining techniques [6].

Recent technologies for monitoring farm animals rely on computer vision and machine learning algorithms [2]. The application of digital image processing and machine learning has facilitated a rise in research focused on avian welfare [7]. The goal of environmental enrichment is to enhance environmental complexity [8] by providing opportunities for animals to perform natural behaviors [9] and change broiler behavior and distribution patterns [10,11,12,13,14,15].

Among all environmental stressors, thermal stress is the most harmful, with great emphasis on animal agriculture [16]. When exposed to heat stress, broilers exhibit behavioral, immunological, and physiological changes that negatively impact their productivity [17]. Husbandry workers routinely observe animal postural patterns to assess thermal comfort and adjust environmental settings or management [18]. Environmental enrichment during early age (1 to 22 days) effectively mitigates fear and thermal stress responses in broiler chickens to unexpected environmental changes without adversely affecting growth performance and stress status [11]. Computer vision has enabled the detection of the impact of environmental alterations (environmental enrichment and temperature) on the locomotion of broiler chickens housed in a controlled climate chamber [19]. Behavioral analysis holds significant potential in developing a remote monitoring system for detecting heat stress in poultry reared in non-enriched environments [20,21,22]. Therefore, behavioral analysis represents a promising and non-invasive tool for estimating the level of animal comfort and welfare.

For an extended period, behavioral responses have predominantly been quantified based on their intensity, frequency, or duration [23]. The gold standard for video analysis in behavioral research has been the implementation of the “scan-sampling” method by researchers [24]. On the other hand, the intricate patterns of behavior encompassing directional, sequential, and temporal organization have been largely disregarded and underestimated [23]. Nevertheless, research interests progressively shift towards analyzing more intricate behavioral patterns [23,25,26,27]. Previous research [28] indicated that the complexity of locomotive sequences diminished under stress, while the complexity of behaviors such as perching, foraging, and resting increased in enriched conditions. Ref. [25] showed that chickens can sustain a consistent behavioral pattern during rearing, characterized by individual rather than flock-based rhythms, as evidenced by their spatial utilization within the barn. Such distinct rhythmic behavior indicates an organism’s health [29]. The existing literature recognizes that stressors can disrupt behavioral rhythms [29,30,31,32,33]. Consequently, detecting recurrent behavioral patterns over time can be crucial for evaluating breeding practices and animal welfare.

The Generalized Sequential Pattern (GSP) algorithm is a data mining method that identifies recurring sequences surpassing a user-specified support threshold, initially developed by [34] to understand customer behavior patterns. The potential of this algorithm for the behavioral profiling of broiler chickens in conditions of thermal comfort and heat stress was investigated by [26,27]. However, its applicability for poultry housed in enriched environments remains unexplored.

The current study serves as a proof of concept aimed at validating the application of the GSP algorithm for identifying temporal correlations between heat stress and the behavior of broiler chickens in enriched environments.

## 2. Materials and Methods

### 2.1. Animals, Housing, and Management

Twenty-one-day-old, mixed-sex Cobb^®^ chicks were sourced from a commercial hatchery and transferred to a climate chamber. They had similar weights and balanced sex distribution. The climatic chamber contained three compartments with an independent climate control system. The compartments and animals were randomly assigned to treatment groups on the first day of housing and divided into two treatments, each comprising ten broilers: one without environmental enrichment and one with environmental enrichment. Experimental testing occurred only at 21 and 22 days of broiler age under thermoneutral and heat stress conditions, respectively. This age marks the onset of heat stress that can impair productive performance (reduced feed intake and weight gain) and adversely challenge animal metabolism and immunity [35,36,37].

Neither compartment contained environmental enrichment objects during the initial acclimatization period (the first three days) in the climate chamber. The broilers were encouraged to consume food and water. From the fourth day onward, the compartment designated as “enriched” was provided with colored plastic rings suspended by a string, a plastic box containing fine sand, and a wooden perch designed to positively stimulate the species’ natural behaviors (perching, pecking, and dust bathing) [38,39,40]. The enrichment objects were rearranged within the system every three days, following the methodology proposed by [10], to promote exploratory behavior and prevent loss of interest by the animals.

Each compartment (measuring 1.6 m × 1.4 m) was equipped with an air conditioner, two dehumidifiers, two heaters, a dimmable LED lamp to control light intensity (lux), and a video camera. Throughout the experimental period, water and commercial feed were provided ad libitum, adhering to the nutritional recommendations suggested by [41]. Each compartment contained a manual tubular feeder and an automatic tubular drinker with height adjustment. Daily morning management included cleaning the floor and the drinker, and feed was weighed and distributed into the feeders by the same personnel. The floor was covered with bedding made of wood shavings (0.05 m). The lighting schedule followed company recommendations, with 24 h light until day 7, followed by an increase of 1 h of darkness every 2 days, so that by day 14, the broilers were subjected to a photoperiod of 20 h of light and 4 h of darkness, from 21:00 h to 01:00 h, until the end of the rearing period at 42 days of age. The adopted illuminance followed the recommendation of 20 lx measured at the height of the birds and was gradually reduced to 5 to 10 lx until the end of the housing [42]. The environmental control center manages each compartment of the controlled environment room using software developed with the Delphi programming language (version 6.0, Borland Software Co., Austin, TX, USA). This software enables the measurement, processing, control, and recording of continuously collected data. The system allows users to monitor real-time temperature, humidity, light intensity, and air renewal rates. The environmental control equipment was automatically activated (turned on and off) according to the temperature and humidity levels set by the user. Relative humidity was programmed to remain at 60% continuously [42], with only air temperature varying during the experimental period. The automated control system was demonstrated to maintain distinct set-point temperatures and required one hour to stabilize at the desired temperature [43]. Figure 1 [19] shows the positioning of all the equipment used in the climate chamber and the technical support room during the experimental period.

During the initial twenty-one days of housing, the broiler chickens were kept in thermoneutral conditions. On the twenty-second day, they were exposed to heat stress (30 °C) for a continuous seven-hour period from 12:00 to 18:00 h, following the standards set by the Cobb breeding manual [42]. The initial phase of the study was not documented; instead, our research focused exclusively on the growth phase for two days as a proof of concept. This investigation recorded behaviors of birds with intact plumage that were yet sufficiently lightweight enough to exhibit natural behaviors. This approach was adopted because older broiler chickens tend to exhibit reduced locomotion, which could compromise the validation of our computational model for behavior sequencing, the central aspect of this study. The data collected were adequate to substantiate the concept. The proof of concept aims to show preliminary evidence demonstrating the feasibility of a novel idea, method, or innovation, thereby establishing a foundation for more extensive development, testing, or comprehensive implementation.

### 2.2. Image Acquisition

Surveillance cameras (Intelbras^®^ VMD 3120 IR, Intelbras Corporation, São José, Santa Catarina, Brazil) were employed to acquire and subsequently analyze animal behavior data. These devices offer a resolution of 976 × 496 (H × V) and are equipped with an infrared feature that activates automatically under low-light conditions. The cameras were mounted on the ceiling at the geometric center of each compartment. Figure 2 shows the unenriched (A) and enriched (B) compartments.

Data collection occurred through video recordings captured between 12:00 and 18:00 h. The recordings were then automatically archived on an NVR video recorder (Intelbras^®^ Multi HD Serie 1000, 1080p, Intelbras Corporation, São José, Santa Catarina, Brazil).

Video image collection was conducted automatically for 48 consecutive hours, with an analysis of a continuous 6-h period from 12:00 to 18:00 for two consecutive days of testing. The final dataset included one behavioral attribute with thirteen possible ones (Table 1), representing fundamental behaviors identified in previous studies [26,27] on the behavior and welfare of broiler chickens in both non-enriched and enriched environments.

The analysis of behavioral time series necessitates the utilization of uninterrupted datasets, as the unique response patterns of each subject may provide novel insights into an animal’s condition [23]. The present study involved video recordings of 30-min durations, focusing on the initial 10 min, segmented into two 5 min intervals for continuous analysis. This research further sought to evaluate the precision of the Generalized Sequential Pattern (GSP) algorithm within this abbreviated timeframe, a context not previously explored in the scientific literature. Previous research [26,27] identified temporal sequences using the GSP algorithm during a continuous 15 and 10 min analysis.

Behavioral transitions were identified when an avian subject exhibited a specific activity for at least 10 s before switching to a distinct behavior [27,39,47]. On each assessment day, under usual and heat stress conditions, we observed 140 behavioral events for each broiler (n = 10), corresponding to uninterrupted behavioral observations. These were segmented into 14 video clips, each 5 min in length. The recorded behaviors were subsequently sequenced and analyzed using Weka^®^ software (version 4.3.0), which executed the sequential pattern mining tasks, such as the Generalized Sequential Pattern (GSP) algorithm. This software is a comprehensive platform combining algorithms from diverse methodologies within the subfield of artificial intelligence, mainly focusing on machine learning applications [48]. Figure 3 illustrates the diagram of activities used in data collection for proof-of-concept validation.

### 2.3. Generalized Sequential Pattern (GSP) Algorithm

Sequential pattern mining is employed for predictive analysis in data mining [49]. The GSP algorithm, created and implemented by [34], is intended to unearth patterns frequently occurring over time within databases, adhering to a user-determined minimum support level. Therefore, supporting a sequential pattern equates to the fraction of data sequences that encapsulate the pattern. However, each item within a sequence pattern must co-occur within a single transaction for the data sequence to validate the pattern [34]. For instance, the algorithm conducts several passes over the dataset. The initial pass assesses the support of each item—that is, it counts the data sequences containing the item and identifies which items are frequent and meet the threshold of minimal support. These items, in turn, each generate a single-element frequent sequence. Following this, each subsequent pass is initiated with a set of seeds, the frequent sequences discovered in the prior pass. This seed set is then employed to cultivate new candidate sequences, potentially frequent, each augmented by one additional item. The support for these candidate sequences is gauged during the pass through the data. At the end of the cycle, the algorithm ascertains which candidate sequences are frequent, establishing them as seeds for the ensuing pass. The algorithm concludes its process when no frequent sequences are detected at the close of a pass or when no new candidate sequences are generated [34,49].

A comprehensive exploration of the fundamental principles of sequence pattern mining is elaborated upon in several studies [34,50,51,52]. By utilizing a user-defined minimum support threshold (minsup), the Generalized Sequential Pattern (GSP) algorithm seeks to identify all subsequences within a sequence database that meet or exceed this threshold [52].

The primary contributions of the present study are to demonstrate that environmental enrichment modifies behavioral patterns, showcasing the potential of the GSP algorithm for monitoring heat stress and suggesting that enrichment may offer improved opportunities for stress adaptation.

The support of a sequence is quantified using Equation (1), which establishes the basis for evaluating the frequency of subsequence occurrences relative to the total dataset.
(1)$$\frac{support\;\left(s\right)=\left|Number\;of\;occurrences\;S\right|}{\left|Total\;of\;sequence\;in\;the\;data\;set\right|}\to \left[0;1\right]$$

Table 2 illustrates data construction on sequential patterns in broiler chickens using the Generalized Sequential Pattern (GSP) algorithm. The Sequence-Id (SID) corresponds to the identification of each sampled chicken (e.g., broiler_1_, broiler_2_. to broiler_n_). The time transaction records observations according to user-defined temporal criteria (1st, 2nd, 3rd, …). Behaviors are denoted by the frequency of their occurrence within the analyzed minutes and constitute the items in the sequence. The number of items within a sequence defines its length. The data prepared for analysis in the data mining platform comprise tuples formatted as <SID, s>, where “SID” is the Sequence-Id and “s” consists of a list of temporally ordered behaviors of broiler chickens.

In the example (Table 2), we have SID (broiler_1_) = <Eat (E), Walk (W), Drink (D)>, which has a size of 3 and means that these behaviors occurred in this chronological order, and SID (broiler_2_) = <Lying down (Ld), Preening (P)> with a size of 2. The minimum support value established as a criterion by the user will determine whether this sequence is frequent or not in the database. Agrawal and Srikant [34] pioneered the development and validation of the GSP algorithm, which was designed to understand customer purchasing behavior in commercial retail. They used a minimum support threshold (minsup) of 5% to find frequent sequential patterns in a specific database. Prior research [26,27] investigating the sequential behavior of broiler chickens in unenriched environments under varying thermal conditions utilized minimum support thresholds of 25% and 40%, respectively, within their Generalized Sequential Pattern (GSP) analyses. The present study used a 20% threshold, the best-performing threshold for evaluating the sequential pattern of broiler chickens housed in an enriched environment. The support value of any sequence reveals how frequent that sequence is.

## 3. Results and Discussion

Table 3 presents the frequent behavioral patterns for broiler chickens aged 21 days (third week) reared in an unenriched and enriched environment under comfortable temperatures. Table 4 inserts the frequent behavioral patterns for broiler chickens at 22 days reared in an unenriched and enriched environment under heat stress. As the MinSupport value is reduced, new sequences are found, and the results (sequences) of the previous supports are excluded from the tables because they are identical subsets at the higher level. This pruning reduces repetitiveness and facilitates the reader’s analysis.

Of all the behavioral possibilities described and considered in the ethogram of Table 1, 4 out of the total 13 behaviors were not observed (frequent) at the 20% support (run, dust bathing, explore, and wing flap). This result may be rooted in some relevant aspects, either in isolation or combination. Factors that explain our results include specificities involving the tool (the operational aspect of the GSP algorithm that requires the behavior frequency to occur in the chronological order in which they appeared) [34,50]; particularities inherent to the behavior itself, such as low frequency (rare events); characteristics related to the individual; and also the bias of chance (“ error” of uncontrolled origin).

The comfort temperature resulted in a more significant number of behavioral patterns compared to heat stress. The number of patterns found between the treatments was similar under thermoneutral conditions (27 non-enriched vs. 29 enriched) and heat stress conditions (7 non-enriched vs. 8 enriched). Sequence sizes ranging from 1 to 4 were identified in both rearing environments under thermoneutrality. The size 4 sequence in the non-enriched environment shows that the broiler performed the behaviors in chronological order according to the analysis criterion (during 5 continuous minutes): <{P,F,P}–1 step {Ld,P}–2 step {Ld,F}–3 step {Ld,P}–4 step>. In the enriched environment, the size 4 sequence involved the same behaviors but in a different chronological order: <{Ld,P}–1 step {Ld,P,F,P}–2 step {Ld,F}–3 step {Ld,F}–4 step>. However, the ambient temperature above the comfort zone restricted the sequence sizes to a maximum of 2. Heat stress drastically reduced the number of pattern sequences of broilers in both tested situations, reducing the level of activity and movement [17,19,20,21,22,26,27,44]. Ref. [28] studied broilers in an enriched environment and established that the fractal complexity of the behavior sequence decreased with increased heat stress due to insufficient energy to perform complex behaviors.

The sequence <{Ld,P}> was the first behavioral pattern found in both treatments (non-enriched and enriched) under the comfort temperature and heat stress only in broilers housed in a non-enriched environment. Branco et al. [26] also observed this same sequential pattern in broilers of the same age (21 days) in non-enriched and thermoneutral situations. The most frequent behavior in the sequential patterns was Ld (lying down), mainly under the comfort temperature. Broilers spend an average of 76–80% of their time lying down [53], which predisposes them to locomotor problems (lesions and fractures) due to poor movement [15,54]. Lameness, long periods of lying down, and other health problems will likely interfere with the broilers’ ability to perform high-priority behaviors [55], such as foraging, comfort, and dust bathing [56]. Preening is a natural broiler behavior, showing they are highly motivated and also indicative of comfort [53,54], although it may be more present in thermal stress environments [26]. Specifically, for this research, comfort behaviors considered were preening (P), dust bathing (Db), shaking feathers (Sf), and stretching (Sf) [55]. Foraging (F) and exploring (Ex) behaviors are classified as exploratory behavior [55], being essential and considered healthy for broilers. Our sequence pattern results identified a high prevalence of” Ld”,” P”, and” F” in the sequence formation in both treatments, indicating strong motivation and priority from the animal’s perspective. Broilers in the comfort temperature have high motivation to perform these last-mentioned behaviors, whose sequences <{Ld,P}>; <{Ld,F}>; <{P,F,P}>; <{Ld,P,F}>; and <{E,W,F}> were the only ones observed in both treatments. All other sequential patterns (comfort and stress) were distinct, suggesting that environmental enrichment alters the behavioral pattern of broilers.

The walking behavior of broilers in thermoneutrality was more directly related to feeding (eating and drinking) and exploration activities (foraging), as observed in the sequences <{E,W,F}>; <{W,F,P}>; <{D,W,E}>; and <{F,P,W,E}>. Broilers explore the environment in search of food and water even under thermal stress situations, as seen in non-enriched housing with <{E,W,P,W,F}> and <{Ld,P,St,Ll}> <{Ld,P,W,D,W}> and enriched housing with <{E,W,F}>; <{Ld,W,D}>; <{E,W,D}>; and <{P,F,St,P}> <{E,W,F}>. In a conventional statistical approach, ref. [57] observed frequencies ≥ 20% for the behaviors of lying down, walking, eating, drinking, and preening, reinforcing from another perspective the probable connection of locomotor activity behavior related to feeding. Broilers housed in non-enriched environments under heat-stress conditions tend to walk and congregate near the drinker to benefit from the microclimate near the water [19,21,58]. However, in the same situations but in enriched environments, they gather around enrichment objects, indicating that environmental enrichment can minimize the negative effect of thermal stress on the broilers [19]. Rufener et al. [25] investigated behavioral time series in laying hens, revealing that the birds exhibited consistent daily results specific to each individual’s movement and location patterns. In other words, the hens can maintain a relatively stable behavioral rhythm under rearing conditions, following an individual rhythm rather than a flock rhythm, which can be reflected in the use of areas in the barn [25].

In the thermally comfortable and enriched environment, the broilers exhibited the sequence <{Ld,P,Sf,P}>, indicating probable positive well-being, as the behavior” Sf” was not observed in any pattern for broilers in a sterile environment. It should be noted that the more frequent a behavior is, the higher the probability of the GSP algorithm finding a pattern according to chronology and threshold criteria.

The most prominent sequential pattern for broilers raised in environments devoid of environmental enrichment when thermally challenged were <{Ld,P,St,Ll}> and <{Ld,P,St,Ll} {Ld,P,W,D,W}>. Specifically for the sequence” St,Ll”, some critical points should be evaluated. Firstly, the behavior” St” is classified as a comfort behavior for broilers and could be used to assess animal welfare [59]. However, measuring its occurrence in isolation may not be the best way to estimate the welfare status of this animal, and it should be correlated with climatic factors. Thus, an individual signature in temporal behavior patterns could provide a new opportunity to assess an animal’s state [23]. Therefore, the chronological context should be specially considered, as the subsequent behavior observed in the sequence of” St” is lying laterally” Ll”, which is a strong indicator of heat stress [26,27]. In other words, chronologically speaking, if the broiler performs” St” and then immediately lies laterally, the goal of this broiler is to increase the body surface area for heat exchange (dissipation) for thermal relief [26,27,60]. Conversely, we did not observe this” St,Ll” pattern for broilers housed in enriched environments, which may suggest that environmental enrichment provides the animal with better opportunities for adaptation to face stress-inducing challenges, such as heat stress, corroborating the findings of [10,19].

However, under heat stress in an enriched environment, there was a tendency for a lower prevalence of the Ld behavior. The sequences observed in broilers housed in enriched environments indicated a more active repertoire involving activities such as the sequences <{Ld,W,D}> (n = 2); <{Ld,P,W,P}> (n = 2); <{E,W,D}> (n = 2); <{P,F,St,P}> (n = 2); <{E,F,E}> (n = 2); and <{P,F,St,P} {E,W,F}>. Various authors reported more significant movement of broilers in environments provided with environmental enrichment [11,15,19,61,62], correlating with a broader range of usual and motivated behaviors [38,63,64] and, therefore, with potentially fewer health and welfare impairments [65,66]. Foremost, we suggest future research encompassing all broiler ages to find differences in all phases of broiler growth.

## 4. Conclusions

Using the GSP algorithm, we identified behavioral sequences performed by broiler chickens at 21 and 22 days of age. Heat stress drastically reduced the complexity of behaviors expressed over time. Environmental enrichment provided greater behavioral diversity and more complex temporal sequences, suggesting improved welfare compared to thermal comfort results. Our study offers valuable insights for estimating the welfare of broiler chickens through the study of temporal behavioral sequences.

## Figures and Tables

**Figure 1 animals-14-02010-f001:**
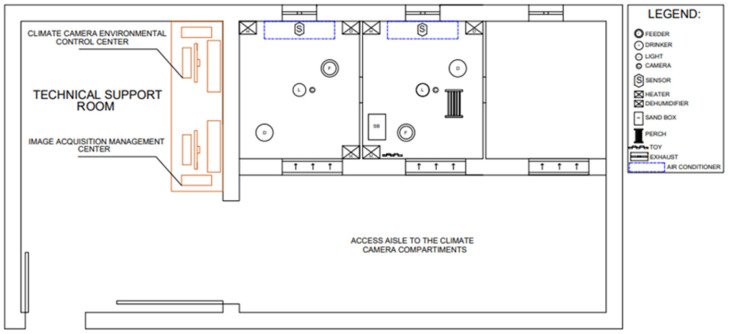
Plan view of the environmental chamber [19]. Reprinted/adapted with permission from Massari et al. [19].

**Figure 2 animals-14-02010-f002:**
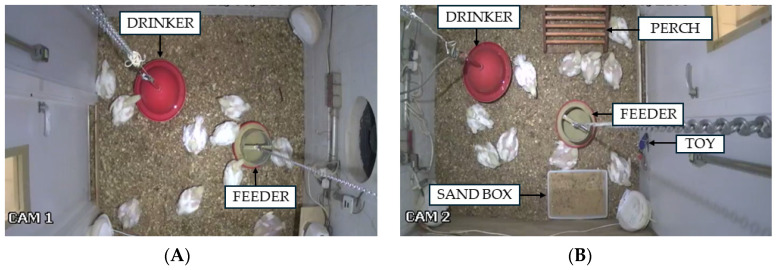
Top view of the unenriched (**A**) and enriched (**B**) compartments. Reprinted/adapted with permission from Ref. [19], 2022, Juliana Maria Massari.

**Figure 3 animals-14-02010-f003:**
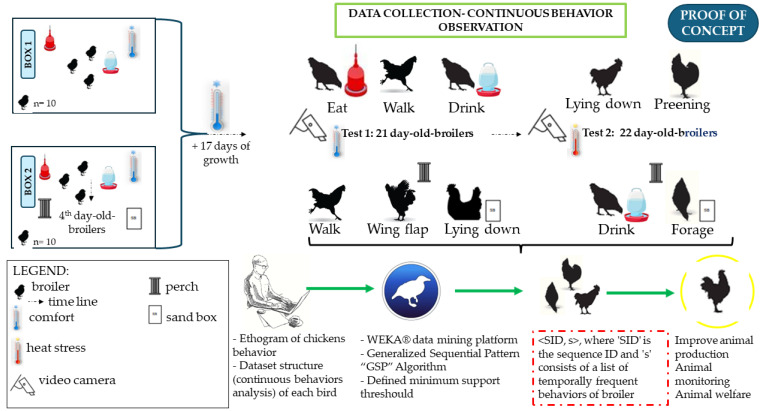
Schematic of the research development for proof-of-concept tests. The hatched line focuses on the proof-of-concept experiment that leads to a sequence of annotated behaviors.

**Table 1 animals-14-02010-t001:** The descriptive ethogram of broiler chicken behaviors applied to compile the database.

Behavior	Acronym	Description	Reference
Lying down	Ld	Broilers with the ventral body region in contact with the ground, knees bent, and with either closed or open eyes (resting).	[44]
Preening	P	Broilers pecking or scratching their feathers in both seated and standing positions.	[45]
Forage	F	Broilers stretch their necks and peck at the substrate on the ground in both sitting and standing positions.	[15]
Walk	W	Broilers walk at least two steps without pecking at the ground.	[45]
Dust bathing	Db	The broiler is lying on the ground, throwing dirt on its back/wings, ruffling, and shaking its feathers.	[46]
Wing flap	Wf	The broiler is repeatedly seen flapping its wings.	[47]
Shake feathers	Sf	The broiler is ruffling and shaking all the feathers of its body.	[44]
Eat	E	The broiler’s head is positioned within the feeder, encompassing slight movements around the perimeter of the feeder.	[45]
Drink	D	The broiler’s beak is in contact with the drinker, including slight movements around the drinker.	[45]
Run	R	The broiler achieves a higher ground speed when the propulsive force is derived from leg action.	[47]
Stretching	St	The broiler stretches one or two wings or legs and returns to its original position.	[15]
Lying laterally	Ll	The broiler is lying on its side with one leg and/or wing stretched out.	[15]
Explore	Ex	The broiler explores the environment by interacting with objects. This is exclusive to the enriched environment, including pecking perch, sandbox, and colorful plastic rings.	[11]

**Table 2 animals-14-02010-t002:** Example of database for analyzing sequential behavioral patterns in broiler chickens.

Database *D*		
Sequence-Id	Transaction Time	Items
Broiler 1	1st	Eat (E)
Broiler 1	2nd	Walk (W)
Broiler 1	3rd	Drink (D)
Broiler 2	1st	Lying down (Ld)
Broiler 2	2nd	Preening (P)

**Table 3 animals-14-02010-t003:** Sequential behavioral patterns of broiler chickens in non-enriched and enriched environments under thermoneutral conditions.

**Broilers Housed in a Non-Enriched Environment**		
**MinSupport**	**Sequence Number**	**Sequence**
1.0, 0.9 and 0.8	1 size 1	<{Ld,P}> (n = 10)
0.7 and 0.6	1 size 1 and 1 size 2	<{Ld,F}> (n = 7); <{Ld,P} {Ld,P}> (n = 7)
0.5	1 size 2	<{Ld,F} {Ld,P}> (n = 5)
0.4	1 size 2	<{Ld,P} {Ld,F}> (n = 4)
0.3	2 size 1 and 1 size 3	<{P,F,P}> (n = 3); <{Ld,P,F}> (n = 3); <{Ld,P} {Ld,P} {Ld,P}> (n = 3)
0.2	4 size 1, 8 size 2, 6 size 3, and 1 size 4	<{E,W,F}> (n = 2); <{W,F,P}> (n = 2); <{D,W,E}> (n = 2); <{F,P,W,E}> (n = 2); <{Ld,P} {Ld,P,F}> (n = 2); <{Ld,P} {D,W,E}> (n = 2); <{P,F,P} {Ld,P}> (n = 2); <{P,F,P} {Ld,F}> (n = 2); <{Ld,F} {Ld,P,F}> (n = 2); <{Ld,F} {D,W,E}> (n = 2); <{Ld,P,F} {Ld,P}> (n = 2); <{W,F,P} {Ld,P}> (n = 2); <{Ld,P} {Ld,P} {Ld,F}> (n = 2); <{Ld,P} {Ld,F} {Ld,P}> (n = 2); <{P,F,P} {Ld,P} {Ld,P}> (n = 2); <{P,F,P} {Ld,P} {Ld,F}> (n = 2); <{Ld,F} {Ld,P} {Ld,P}> (n = 2); <{Ld,F} {Ld,P} {Ld,P,F}> (n = 2); <{P,F,P} {Ld,P} {Ld,F} {Ld,P}> (n = 2)
**Broilers Housed in an Enriched Environment**		
**MinSupport**	**Sequence Number**	**Sequence**
1.0 to 0.8	None	
0.7 and 0.6	1 size 1	<{Ld,P}> (n = 7)
0.5	2 size 1	<{Ld,P,F,P}> (n = 5); <{Ld,P,F}> (n = 5)
0.4	1 size 1	<{F,P}> (n = 4)
0.3	3 size 1 and 2 size 2	<{P,F,P}> (n = 3); <{P,Ld}> (n = 3); <{Ld,F,P}> (n = 3); <{Ld,P} {Ld,P,F,P}> (n = 3); <{Ld,P,F,P} {Ld,P,F,P}> (n = 3)
0.2	4 size 1, 11 size 2, 4 size 3, and 1 size 4	<{Ld,F,P,F}> (n = 2); <{Ld,F}> (n = 2); <{Ld,P,Sf,P}> (n = 2); <{E,W,F}> (n = 2); <{Ld,P} {Ld,F}> (n = 2); <{Ld,P} {Ld,F,P}> (n = 2); <{F,P} {Ld,P,F,P}> (n = 2); <{Ld,P,F,P} {Ld,F}> (n = 2); <{P,F,P} {Ld,P}> (n = 2); <{P,F,P} {Ld,P,F,P}> (n = 2); <{Ld,F} {Ld,F}> (n = 2); <{P,Ld} {Ld,P}>; <{P,Ld} {Ld, P,F}>; <{E,W,F} {F,P}> (n = 2); <{Ld,P,F} {Ld,P}> (n = 2); <{Ld,P} {Ld,F} {Ld,F}> (n = 2); <{Ld,P} {Ld,P,F,P} {Ld,P,F,P}> (n = 2); <{Ld,P} {Ld,P,F,P} {Ld,F}> (n = 2); <{Ld,P,F,P} {Ld,F} {Ld,F}> (n = 2); <{Ld,P} {Ld,P,F,P} {Ld,F} {Ld,F}> (n = 2)

n = Number of broilers performing the described behavior with a support value of 20%. Legend behaviors: Ld = lying down; P = preening; F = forage; E = eat; D = drink; W = walk; Sf = shake feathers.

**Table 4 animals-14-02010-t004:** The sequential behavioral patterns of broiler chickens housed in a non-enriched and enriched environment under heat stress conditions.

**Broilers Housed in a Non-Enriched Environment**		
**MinSupport**	**Sequence Number**	**Sequence**
1.0 to 0.5	None	
0.4	1 size 1	<{Ld,P,W,D,W}> (n = 4)
0.3	2 size 1	<{Ld,P}> (n = 3); <{Ld,F,P}> (n = 3)
0.2	3 size 1 and 1 size 2	<{Ld,P,F}> (n = 2); <{Ld,P,St,Ll}> (n = 2); <{E,W,P,W,F}> (n = 2); <{Ld,P,St,Ll} {Ld,P,W,D,W}> (n = 2)
**Broilers Housed in an Enriched Environment**		
**MinSupport**	**Sequence Number**	**Sequence**
1.0 to 0.4	None	
0.3	1 size 1	<{E,W,F}> (n = 3)
0.2	6 size 1 and 1 size 2	<{Ld,F}> (n = 2); <{Ld,W,D}> (n = 2); <{Ld,P,W,P}> (n = 2); <{E,W,D}> (n = 2); <{P,F,St,P}> (n = 2); <{E,F,E}> (n = 2); <{P,F,St,P} {E,W,F}> (n = 2)

n = Number of broilers performing the described behavior with a support value of 20%. Legend behaviors: Ld = lying down; P = preening; F = forage; E = eat; D = drink; W = walk; St = stretching; Ll = lying laterally.

## Data Availability

Data will be available upon request.

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
