# Peer review of "Sequential Behavior of Broiler Chickens in Enriched Environments under Varying Thermal Conditions Using the Generalized Sequential Pattern Algorithm: A Proof of Concept"

_animals, 2024, doi:10.3390/ani14132010_

Round 1

Reviewer 1 Report

Comments and Suggestions for Authors

The aim of the paper is to validate the application of the generalized sequential pattern (GSP) algorithm to identify temporal correlations between heat stress and the behavior of broiler chickens in enriched environments. The main contributions include demonstrating that environmental enrichment alters behavioral patterns, highlighting the potential of the GSP algorithm for monitoring hear stress, and suggesting that enrichment may provide better opportunities for adaptation to stress.

Line 193: (Table 1) the description of behaviors in the ethogram is clear, but additional references to previous studies that have used similar ethograms would strengthen the justification for these choices. It is recommended to cite at least one study with a similar ethogram, not just to define the behaviors.

Line 197: stylistically, it could be clearer.

Suggested revision; “This study involved video recordings of 30-minute durations, focusing on the initial 10 minutes, segmented into two 5-minute intervals for continuous analysis.

Line 243: (Figure 3) the schematic of the methodology is helpful, but it would benefit from a more detailed legend explaining each step of the process.

Abstract:

It provides a concise summary of the study, including its aim, methodology, main findings, and conclusions. It is well structured and gives a clear overview of what the paper covers.

Introduction:

Effectively sets the context for the study, explaining the relevance for broiler behavior and environmental enrichment.

I suggest highlighting the unique contributions of your study.

Materials and methods

This section is detailed, describing the experimental setup, data collection, and analytical techniques used. It provides sufficient information for reproducibility.

Maybe you can provide more detailed descriptions of the environmental conditions such as temperature and humidity control settings, and how these were maintained throughout the study.

Results and discussion

This section presents the findings clearly, with appropriate use of tables and figures.

Suggest directions for future research to highlight the study´s contributions and potential impact on the field.

Conclusion

It summarizes the main findings and their implications. It reinforces the study´s significance and suggest practical applications.

Author Response

Dear Reviewer 1, we appreciate your corrections and suggestions that improved considerably the manuscript, and we answer one by one as follows:

Line 193: (Table 1) the description of behaviors in the ethogram is clear, but additional references to previous studies that have used similar ethograms would strengthen the justification for these choices. It is recommended to cite at least one study with a similar ethogram, not just to define the behaviors.

Answer: Two studies were cited having similar ethograms

Line 197: stylistically, it could be clearer.

Suggested revision; “This study involved video recordings of 30-minute durations, focusing on the initial 10 minutes, segmented into two 5-minute intervals for continuous analysis.

Answer: We accept the suggestion. 

Line 243: (Figure 3) the schematic of the methodology is helpful, but it would benefit from a more detailed legend explaining each step of the process.

Answer: The Figure was improved as well the legend.

Introduction:

Effectively sets the context for the study, explaining the relevance for broiler behavior and environmental enrichment.

I suggest highlighting the unique contributions of your study.

Answer: The contributions of our study were highlighted.

Materials and methods

This section is detailed, describing the experimental setup, data collection, and analytical techniques used. It provides sufficient information for reproducibility.

Maybe you can provide more detailed descriptions of the environmental conditions such as temperature and humidity control settings, and how these were maintained throughout the study.

Answer: We provide more detailed descriptions of the environmental conditions, such as temperature and humidity control settings, and how these were maintained throughout the study

Results and discussion

This section presents the findings clearly, with appropriate use of tables and figures.

Suggest directions for future research to highlight the study´s contributions and potential impact on the field.

Answer: We suggest future research using all broiler ages to see differences in all phases of broiler growth.

Reviewer 2 Report

Comments and Suggestions for Authors

Review of manuscript animals-3074262: Sequential Behavior of Broiler Chickens in Enriched Environ-2 ments Under Varying Thermal Conditions Using the General-3 ized Sequential Pattern Algorithm: A Proof of Concept.

General comments

The study aimed to evaluate Sequential Behavior of Broiler Chickens in Enriched Environments under varying thermal conditions using the Generalized Sequential Pattern Algorithm (GPS). The objective of this study is pertinent and current.

The manuscript is well-written but requires attentive review as details/data have been missed or have not been properly formatted.

There is a need for a major review throughout the manuscript. In the next session, there’s a list of suggestions for the authors to improve the manuscript.

The major areas that need to be addressed as a priority are Material and Method:

2.1. Animal, housing, and management

1)                  The authors should better describe the temperature and humidity conditions in which observations are carried out in both thermoneutrality and heat stress.

2)                  The authors should provide a better description of the experiment. The behavior of broiler chickens in the initial phase is different from the growing phase. What phase are we in? Discrepancies between the text and Figure 3.

Initial phase 1 to 21-day-old broilers, and the growing phase broilers older than 21 d. In order to reduce the variability among studies and because the thermoneutral temperatures varied, the equations were created using the difference in the heat stress temperature related to the upper critical

2.2 Image acquisition

“Data collection occurred through video recordings captured between 12:00 and 18:00h. The recordings were then automatically archived on an NVR video recorder (Intelbras® 186 Multi HD Serie 1000, 1080p, Intelbras Corporation, São José, Santa Catarina, Brazil). Video image collection was conducted automatically for 48 consecutive hours, with an analysis of a continuous seven-hour period from 12:00 to 18:00 for two consecutive days of testing. (line 185-188). What video recordings were captured? At 12:00 and 18:00h or 48 consecutive hours? 

Anaylized over a continuous seven-hour period from 12:00 to 18:00. This interval is six-hour period.

2.3 Generalized Sequential Pattern (GPS) Algorithm.

A few detailed methodology that don't facilitate study replicability

Specific comments:

Line 276: Table 2:  Items: preening vs prenning 

Line292:  Table 4 vs Table 5

Line 278-280: “Table 3 presents the frequent behavioral patterns for broiler chickens aged 21 days 278 (third week) reared in an unenriched and enriched environment under comfortable temperaturas” This text are not correct. The broiler aged 41 day old when image acquisiton and analized. Please correct.

Line 280-281:” Table 4 inserts the frequent behavioral patterns for broiler chickens at 22 days reared in an unenriched and enriched environment under heat stress.”

This is not correct. Heat stress conditions are one day, not 22 days. They are measured on day 22 but the days before the broiler chickens are in thermoneutral conditions. Please correct.

Line 390-391: “Using the GSP algorithm, we identified behavioral sequences performed by broiler chickens at 21 and 22 days of age”. This are not true.  The broiler was 41 and 42 days of age.

Comments on the Quality of English Language

English language fine. No issues detected

Author Response

Dear Reviewer 2, we appreciate your corrections and suggestions that improved considerably the manuscript, and we answer one by one as follows:

2.1. Animal, housing, and management

The authors should better describe the temperature and humidity conditions in which observations are carried out in both thermoneutrality and heat stress.

Answer: We provide more detailed descriptions of the environmental conditions such as temperature and humidity, those control settings, and how these were maintained throughout the study

The authors should provide a better description of the experiment. The behavior of broiler chickens in the initial phase is different from the growing phase. What phase are we in? Discrepancies between the text and Figure 3.

Answer: The initial phase of the study was not documented; instead, our research focused exclusively on the growth phase for two days as a proof of concept. This investigation recorded behaviors of birds with intact plumage that were yet sufficiently lightweight to exhibit natural behaviors. This approach was adopted as older broiler chickens tend to reduce locomotion, which could compromise the validation of our computational model for behavior sequencing, the central aspect of this study. The data collected were adequate to substantiate the concept. The proof of concept aims to show preliminary evidence demonstrating the feasibility of a novel idea, method, or innovation, thereby establishing a foundation for more extensive development, testing, or comprehensive implementation. 

Initial phase 1 to 21-day-old broilers, and the growing phase broilers older than 21 d. In order to reduce the variability among studies and because the thermoneutral temperatures varied, the equations were created using the difference in the heat stress temperature related to the upper critical

Answer: We agree; however, we did not study the initial and the end phases. The initial phase of the study was not documented; instead, our research focused exclusively on the growth phase for two days as a proof of concept. This investigation recorded behaviors of birds with intact plumage that were yet sufficiently lightweight to exhibit natural behaviors. This approach was adopted as older broiler chickens tend to reduce locomotion, which could compromise the validation of our computational model for behavior sequencing, the central aspect of this study. The data collected were adequate to substantiate the concept. The proof of concept aims to show preliminary evidence demonstrating the feasibility of a novel idea, method, or innovation, thereby establishing a foundation for more extensive development, testing, or comprehensive implementation.

2.2 Image acquisition 

"Data collection occurred through video recordings captured between 12:00 and 18:00h. The recordings were then automatically archived on an NVR video recorder (Intelbras® 186 Multi HD Serie 1000, 1080p, Intelbras Corporation, São José, Santa Catarina, Brazil). Video image collection was conducted automatically for 48 consecutive hours, with an analysis of a continuous seven-hour period from 12:00 to 18:00 for two consecutive days of testing. (line 185-188). What video recordings were captured? At 12:00 and 18:00h or 48 consecutive hours? 

Anaylized over a continuous seven-hour period from 12:00 to 18:00. This interval is a six-hour period.

Answer: We provide more detailed descriptions of the environmental conditions, such as temperature and humidity control settings, and how these were maintained throughout the study

2.3 Generalized Sequential Pattern (GPS) Algorithm.

A few detailed methodology that don't facilitate study replicability

Answer: This is a proof of concept tested so that further studies involving behavioral patterns, enrichment, and environmental temperature can be carried out. But we improve the methodology explaining it.

Line 276: Table 2:  Items: preening vs prenning 

Answer: We better explain Table 2.

Line292:  Table 4 vs Table 5

Answer: Corrected from Table 5 to 4.

Line 278-280: "Table 3 presents the frequent behavioral patterns for broiler chickens aged 21 days 278 (third week) reared in an unenri ched and enriched environment under comfortable temperaturas" This text are not correct. The broiler aged 41 day old when image acquisiton and analized. Please correct.

Answer: We only analyze broilers' behavior at 21 and 22 days old, aiming to provide proof of the concept.

Line 280-281:" Table 4 inserts the frequent behavioral patterns for broiler chickens at 22 days reared in an unenriched and enriched environment under heat stress."

Answer: Corrected: at day 22 of age.

This is not correct. Heat stress conditions are one day, not 22 days. They are measured on day 22 but the days before the broiler chickens are in thermoneutral conditions. Please correct.

Answer: Corrected at day 22 of age.

Line 390-391: "Using the GSP algorithm, we identified behavioral sequences performed by broiler chickens at 21 and 22 days of age". This are not true.  The broiler was 41 and 42 days of age.

Answer: This is true; we only analyze broilers that are 21 and 22 days old.

Reviewer 3 Report

Comments and Suggestions for Authors

In this paper, authors investigated the behavior sequences in broiler chickens housed in enriched environments subjected to thermal comfort and heat stress using the Generalized Sequential Pattern (GSP) data mining algorithm, which is easy to understand. Here are some questions that the authors have to address.

1.Lines 157-158: the temperature and humidity levels set by the user, who is the user? And how to set?

2.Line 165: they were exposed to heat stress (30°C), why choose this temperature? Please explain clearly.

3.Line 278 and Line 279: the frequent behavioral patterns for broiler chickens aged 21 days and at 22 days? Why choose this day? Please explain clearly.

4.Line 278: Table 3 only showed the sequence behavioral patterns for broiler chickens aged 21 days and at 22 days, what is the performance of the other days? Could we establish a predictive model?

 5.Line: Heat stress drastically reduced the complexity of behaviors expressed over time. We cannot conclude from the paragraph. Please provide a detailed introduction and some tables of the results.

Comments on the Quality of English Language

Minor editing of English language required.

Author Response

Dear Reviewer 3, we appreciate your corrections and suggestions, which improved the manuscript substantially, and we answer one by one as follows:

1.Lines 157-158: the temperature and humidity levels set by the user, who is the user? And how to set?

Answer: We provide more detailed descriptions of the environmental conditions such as temperature and humidity, those control settings, and how these were maintained throughout the study

2.Line 165: they were exposed to heat stress (30°C), why choose this temperature? Please explain clearly.

Answer: 30°C was choose because is a heat stress temperature for birds with 21 and 22 days old according to Cobb breeding manual (Lines 170 to 173)

3.Line 278 and Line 279: the frequent behavioral patterns for broiler chickens aged 21 days and at 22 days? Why choose this day? Please explain clearly.

Answer: The initial phase of the study was not documented; instead, our research focused exclusively on the growth phase for two days as a proof of concept. This investigation recorded behaviors of birds with intact plumage that were yet sufficiently lightweight to exhibit natural behaviors. This approach was adopted as older broiler chickens tend to reduce locomotion, which could compromise the validation of our computational model for behavior sequencing, the central aspect of this study. The data collected were adequate to substantiate the concept. The proof of concept aims to show preliminary evidence demonstrating the feasibility of a novel idea, method, or innovation, thereby establishing a foundation for more extensive development, testing, or comprehensive implementation.

4.Line 278: Table 3 only showed the sequence behavioral patterns for broiler chickens aged 21 days and at 22 days, what is the performance of the other days? Could we establish a predictive model?

Answer: The initial phase of the study was not documented; instead, our research focused exclusively on the growth phase for two days as a proof of concept. This investigation recorded behaviors of birds with intact plumage that were yet sufficiently lightweight to exhibit natural behaviors. This approach was adopted as older broiler chickens tend to reduce locomotion, which could compromise the validation of our computational model for behavior sequencing, the central aspect of this study. The data collected were adequate to substantiate the concept. The proof of concept aims to show preliminary evidence demonstrating the feasibility of a novel idea, method, or innovation, thereby establishing a foundation for more extensive development, testing, or comprehensive implementation.

5.Line: Heat stress drastically reduced the complexity of behaviors expressed over time. We cannot conclude from the paragraph. Please provide a detailed introduction and some tables of the results.

Answer: We corrected it, we can only conclude for 21 and 22 days old birds.

Round 2

Reviewer 1 Report

Comments and Suggestions for Authors

Your diligent work and attention to detail have significantly enhanced the quality and clarity of the paper. The revised manuscript is now more robust and comprehensible.

Author Response

Dear Reviewer 3, 

Your diligent work and attention to detail have significantly enhanced the quality and clarity of the paper. The revised manuscript is now more robust and comprehensible.

Answer:  Thank you for your your corrections and suggestions that improved the manuscript considerably.

Kind Regards,

Daniella Jorge de Moura

Reviewer 2 Report

Comments and Suggestions for Authors

Review 2 of manuscript animals-3074262

Dear author: most of the suggestions have been solved but I still don't understand some concepts.

I don't understand that the chicks are 21 and 22 days old.

In the text it reads (line 124-  )“Twenty-day-old, mixed-sex Cobb® chicks were sourced from a commercial hatchery  and transferred to a climate chamber”, and in line 130  it read “Experimental testing occurred at 21 and 22 days of age….”  and in lines (134 -141)Neither compartment contained environmental enrichment objects during the initial acclimatization period (the first three days) in the climate chamber. The broilers were encouraged to consume food and water. From the fourth day onward, the compartment designated as "enriched" was provided with colored plastic rings suspended by a string, a plastic box containing fine sand, and a wooden perch designed to positively stimulate the species' natural behaviors (perching, pecking, and dust bathing) ([38-40]. The enrichment objects were rearranged within the system every three days, following the methodology  proposed by [10], to promote exploratory behavior and prevent loss of interest by the animals.

How can the experiment be carried out at 21 and 22 days of age if the broiler chickens are already purchased at 20 days of age?

In line 124:  It should read: One-day-old instead of twenty-day-old.  Would this be correct?

Comments on the Quality of English Language

 English language fine. No issues detected.

Author Response

Reviewer 2

Dear Reviewer 2, we appreciate your corrections and suggestions that improved considerably the manuscript, and we answer one by one as follows:

I don't understand that the chicks are 21 and 22 days old.

In the text it reads (line 124-  )“Twenty-day-old, mixed-sex Cobb® chicks were sourced from a commercial hatchery  and transferred to a climate chamber”, and in line 130  it read “Experimental testing occurred at 21 and 22 days of age….”  and in lines (134 -141)Neither compartment contained environmental enrichment objects during the initial acclimatization period (the first three days) in the climate chamber. The broilers were encouraged to consume food and water. From the fourth day onward, the compartment designated as "enriched" was provided with colored plastic rings suspended by a string, a plastic box containing fine sand, and a wooden perch designed to positively stimulate the species' natural behaviors (perching, pecking, and dust bathing) ([38-40]. The enrichment objects were rearranged within the system every three days, following the methodology proposed by [10], to promote exploratory behavior and prevent loss of interest by the animals.

How can the experiment be carried out at 21 and 22 days of age if the broiler chickens are already purchased at 20 days of age?

In line 124:  It should read: One-day-old instead of twenty-day-old.  Would this be correct?

Answer: Yes we should read One-day-old instead of twenty-day-old. We correct all the text to be more clearly at the methodology. As we said before, the initial phase of the study was not documented; instead, our research focused exclusively on the growth phase for two days as a proof of concept. This investigation recorded behaviors of birds with intact plumage that were yet sufficiently lightweight to exhibit natural behaviors. This approach was adopted as older broiler chickens tend to reduce locomotion, which could compromise the validation of our computational model for behavior sequencing, the central aspect of this study. The data collected were adequate to substantiate the concept. The proof of concept aims to show preliminary evidence demonstrating the feasibility of a novel idea, method, or innovation, thereby establishing a foundation for more extensive development, testing, or comprehensive implementation.

Kind Regards,

Daniella Jorge de Moura